# Research on Deep Defect Detection Method of Cable Lead Sealing Based on Improved Pulsed Eddy Current Excitation

**Qianqiu Shao \*, Songhai Fan and Fenglian Liu**

Power Transmission and Transformation Technology Center, State Grid Sichuan Electric Power Research Institute, Chengdu 610095, China
\* Correspondence: shaoqq2011@sc.sgcc.com.cn

**Abstract:** In order to reduce power failures caused by lead sealing defects, it is necessary to carry out nondestructive testing of cable lead sealings. However, previous studies have focused on the detection of surface and near-surface defects of lead sealings. Thus, an improved pulsed eddy current detection (IPECD) method is introduced to detect the deep defects of cable lead sealings (with depths ranging from 6 to 12 mm), and the frequency range selection principle and the optimization method of initial phase angles of different frequency components of IPECD, used to maximize the peak value of the excitation signal, are first explained in detail. Then, the detection sensitivities of the deep defects before and after the optimization are compared and analyzed based on a simulation. Finally, using the IPECD method, experiments are conducted to study the effects of the defect depth on features of the lift-off point of intersection and the zero-crossing time, enhancing the foundation for the prediction or rapid detection of the depth of lead sealing defects.

**Keywords:** cable lead sealing; nondestructive testing; deep defects; improved pulsed eddy current; zero-crossing time; lift-off point of intersection

## 1. Introduction

The lead sealing of a high-voltage power cable has important sealing and waterproofing effects on the cable terminal and intermediate connection, and it ensures that the metal sheath of the cable connects with other electrical equipment and is well grounded [1]. Once the lead sealing of a cable becomes defective under the action of external force, air and water ingression occurs at the cable joints, and this may cause further accidents, such as breakdown and explosions in serious cases. In 2017 and 2019, power failure accidents happened in the Zhejiang power grid of China due to the lead sealing cracking of 110 kV and 220 kV high-voltage cables, which seriously affected the safe and stable transmission of power loads [2]. Hence, in order to improve the monitoring level of lead sealing quality and reduce the accident rates of power failure caused by lead sealing defects, it is necessary to carry out nondestructive testing and an evaluation of the cable lead sealings.

At present, for the defects of cable lead sealing, the currently used nondestructive testing methods include ultrasonic testing, eddy current testing, radiographic testing and loop resistance testing. However, ultrasonic testing, radiographic testing and loop resistance testing are limited by the structure and material of the lead sealing, and they cannot accurately discriminate the defects in a cable lead sealing [2–4]. Moreover, the eddy current testing method, which has the advantages of high sensitivity and non-contact, has been proven to be effective in detecting the defects of lead sealings [2,5–9]. To ensure the mechanical strength of a lead sealing, the thickness of the cable lead sealing should not be less than 12 mm, which is stipulated in the power industry standard DL/T 344-2010 [10].

However, previous studies have focused on the detection of surface and near-surface defects of lead sealings using commercial eddy current testing equipment, and they have seldom researched the detection methods and eddy current signal characteristics of deep

defects in lead sealings [11]. Moreover, the pulsed eddy current testing (PECT) method has been proven to be a promising nondestructive testing technique. However, during the detection process, energy wasting is an unavoidable disadvantage due to its wide frequency band of the PECT signal in the frequency domain [12]. Although researchers have proposed a frequency-band-selecting pulsed eddy current testing (FSPECT) method based on defect depths [13], it is difficult to maximize the peak value of the excitation signal under the same input power when not adjusting the initial phase angles of the different frequency components, which is not conducive to detect the defects inside lead sealings or between the lead sealing and the aluminum sheath under the condition of heavy wall thickness.

Consequently, the objective of this work is to propose an improved pulsed eddy current detection (IPECD) method and to study the features of IPECD signals in the time domain analysis when detecting the defects of lead sealings ranging from 6 to 12 mm. Firstly, the frequency range selection principle and the optimization method of initial phase angles of different frequency components are illustrated. Secondly, the signal characteristics and sensitivities before and after the optimization of the initial phase angles are analyzed via simulations. Finally, the effects of the defect depth on features of the lift-off point of intersection (LOI) and zero-crossing time (ZCT) are explored via experiments, which enhances the accuracy and reliability of IPECD method for the evaluation of defects.

## 2. Basic Principle of the IPECD Method

The most important advantage that the IPECD method has over the traditional FSPECT method is the ability to optimize the initial phase angles of the different frequency components of the excitation signal, ensuring the maximum peak value of the excitation signal under the same input power and improving the signal-to-noise ratio of the detection signal. The specific steps for the frequency range selection principle and the optimization method of the initial phase angles of the different frequency components of IPECD are as follows:

Firstly, as shown in Figure 1, the defect depths of the lead sealing samples range from $D_2$ to $D_1$. According to the material parameters of the samples and the calculation formula of the eddy current skin depth, the effective frequency range $[f_1, f_2]$ of the excitation signal can be deduced as follows [14]:

$$\begin{cases} f_1 = \frac{1}{\pi \mu_0 \mu_r \sigma D_1^2} \\ f_2 = \frac{1}{\pi \mu_0 \mu_r \sigma D_2^2} \end{cases} \tag{1}$$

where $\mu_0$ is the permeability of the free space; $\mu_r$ is the relative permeability of the lead sealing, which has a value of 1; and $\sigma$ is the conductivity of the lead sealing, which has a value of 4.67 MS/m. Moreover, the frequency range of the excitation signal $[f_1, f_2]$ should be selected as [300 Hz, 1500 Hz], in which 300 Hz and 1500 Hz correspond to the frequencies when detecting defects with depths of 12 mm and 6 mm, respectively.

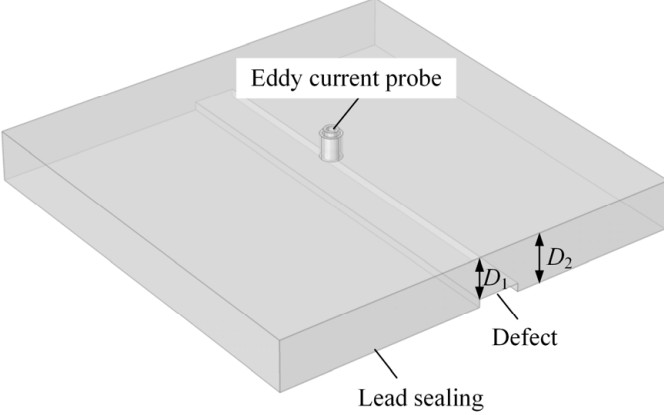

**Figure 1.** Schematic diagram of lead sealing samples with artificial defects.

Secondly, on the basis of the calculated frequency range in Step 1, $[f_1, f_2]$ can be divided into $N - 1$ equal frequency bands, and the $i$th selected frequency can be expressed as

$$f_i = f_1 + \frac{f_2 - f_1}{N - 1}(i - 1), \; i = 1, 2, \cdots, N \tag{2}$$

Assuming that the excitation signal is formed by the superposition of the $N$ sine-wave components of frequency $f_i$ ($i = 1, 2, \ldots, N$), the time domain expression of the excitation signal $I(t)$ before optimization is as follows:

$$I(t) = \sum_{i=1}^{N} A_i \sin(2\pi f_i t + \theta_i) \tag{3}$$

where $A_i$ is the amplitude of the $i$th sine-wave component, and $\theta_i$ is the initial phase angle of the $i$th sine-wave component.

Thirdly, in the process of optimizing the amplitude of the excitation signal $I(t)$, the amplitude and the frequency of each harmonic frequency component are set as known quantities, and the parameters to be optimized are the initial phase angle of each harmonic frequency component. Moreover, the optimization objective is to maximize the maximum amplitude of the excitation signal $I(t)$; thus, the fitness function is defined as follows [15]:

$$f(\theta_1, \theta_2, \cdots, \theta_N) = \max \left\{ \text{abs}[\sum_{i=1}^{N} A_i \sin(2\pi f_i t + \theta_i)] \right\} \tag{4}$$

For the fitness function shown in Equation (4), in the optimization calculation process, the initial phase angles $\theta_i$ ($i = 1, 2, \ldots, N$) of the different frequency components of the excitation signal are firstly set to 0 rad, and then the optimal initial phase angles $\theta_{io}$ ($i = 1, 2, \ldots, N$) of the different frequency components of the excitation signal are calculated using the software named 1stopt with a genetic algorithm.

Fourthly, without the loss of generality, $N$ is set to be 5 in this work, and the optimized initial phase angles corresponding to the excitation signal components with frequencies of 300 Hz, 600 Hz, 900 Hz, 1200 Hz and 1500 Hz are $\theta_{1o} = 0.947$ rad, $\theta_{2o} = 0.324$ rad, $\theta_{3o} = 5.98$ rad, $\theta_{4o} = 5.36$ rad, and $\theta_{5o} = 4.74$ rad, respectively. Moreover, the time domain expression of the excitation signal $I(t)$ after optimization is as follows:

$$I_o(t) = 5 \sum \begin{array}{l} [\sin(600\pi t + 0.947) + \sin(1200\pi t + 0.324) + \sin(1800\pi t + 5.98) + \\ \sin(2400\pi t + 5.36) + \sin(3000\pi t + 4.74)] \end{array} \tag{5}$$

The waveforms of the excitation signals of the different initial phases are shown in Figure 2. It can be clearly concluded that, when the initial phases of each frequency component are the optimized phase, the worst phase and 0, the maximum amplitudes of the excitation signal are 10 A, 4.68 A and 7.92 A, respectively. Moreover, it can be concluded that the IPECD method using the optimized initial phases has a higher sensitivity for the detection of lead sealing defects ranging from 6 mm to 12 mm than the FSPECT method using the initial phases of zero.

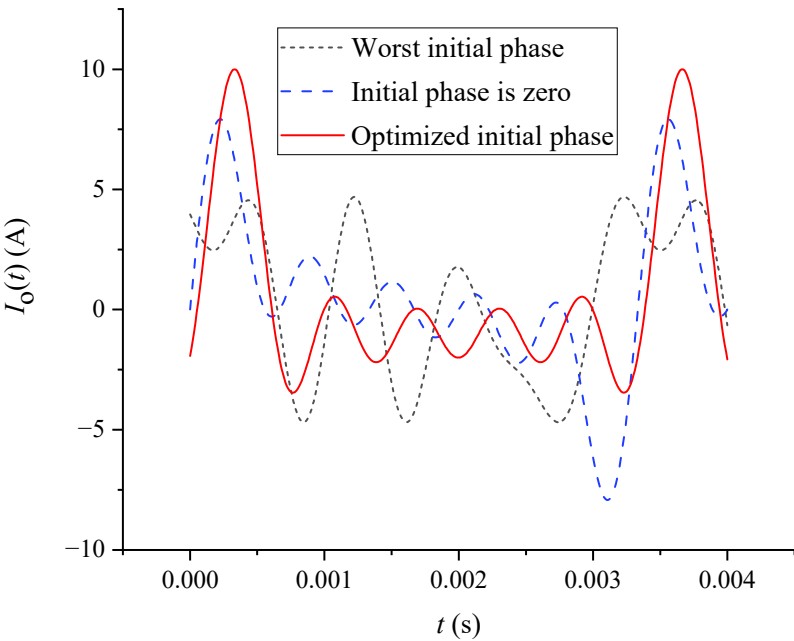

**Figure 2.** Excitation signals with different initial phases.

### 3. Numerical Simulation and Result Analysis

In order to make a comparison of the proposed IPECD method and the traditional FSPECT method, a simulation is performed based on the reduced vector potential method (Ar) to calculate the detection signal strength before and after the optimization of the excitation signal phase angles.

*3.1. Numerical Model*

In this work, COMSOL Multiphysics (Version 5.5, COMSOL AB, Stockholm, Swedeb) is used to establish a two-dimensional axisymmetric finite element simulation model for the eddy current testing of deep defects in lead sealings. The 'magnetic field' in the AC/DC module is selected as the physical field, and the external circuit is coupled to solve the distributions of the magnetic induction intensity in the time domain.

The simulated geometric model is shown in Figure 3. Lead tin alloy plates are employed as samples, and they are 100 mm in diameter, 20 mm in height, 4.67 MS/m in electrical conductivity, 1 in relative permeability and 1 in relative dielectric constant. The samples have bottom defects with 20 mm constant widths and depths ranging from 6 mm to 12 mm. There are seven defects with different depths, and they are all groove defects across the bottom sides of the plates. Moreover, the relative dielectric constant, relative permeability and electrical conductivity of the air domain are set to 1, 1 and 0.01 S/m, respectively. The diameter and the height of the air domain are set to 150 mm and 60 mm, respectively, and the magnetic insulation boundary condition is used in the simulation. The probe consists of an excitation coil and a magnetic field sensor fixed at the bottom center of the excitation coil to measure the vertical magnetic field, while the reference signal is the one obtained for the defect-free specimen. Moreover, the inner/outer diameter, height, wire diameter, turns and lift-off height of the excitation coil are set to 6/10 mm, 5 mm, 0.1 mm, 300 and 0.5 mm, respectively, and their positions are placed above the centers of the defects.

The mesh resolution and mesh element quality are important aspects to consider when validating a model. The mesh parameters in this model are shown in Table 1. The grid number is set to 20,666 in this model, the average element quality is 0.9491, and the minimum element quality is 0.5778, which means a good mesh.

In the simulations, to research the influence of the initial phase of the excitation signal on the detection sensitivity, we load the excitation signals with different initial phases as

shown in Figure 2 on the excitation coil and compare the differential signals of the magnetic induction intensities under different excitation signals.

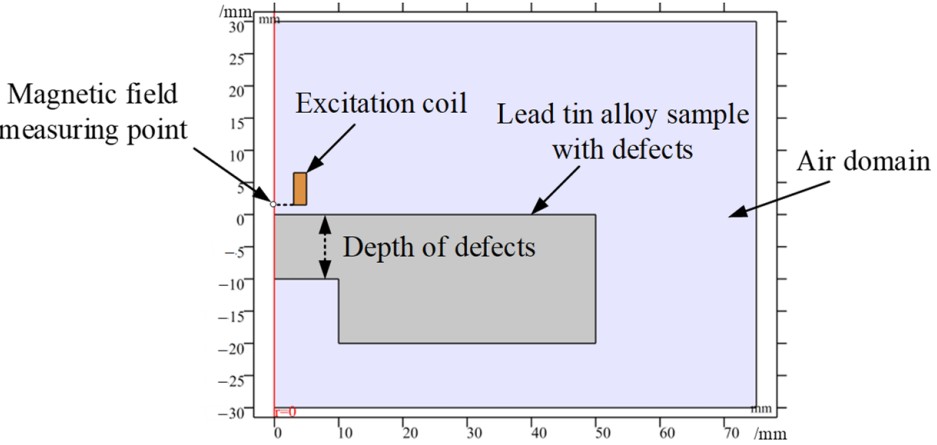

**Figure 3.** Simulation geometry model.

**Table 1.** Table of mesh parameters.

| Mesh Parameters | Values |
| --- | --- |
| Triangular elements | 20,666 |
| Edge elements | 561 |
| Minimum element quality | 0.5778 |
| Average element quality | 0.9491 |

*3.2. Analysis of Simulation Results*

In the simulation, the energies of the excitation signals with different initial phases are the same, which makes the simulation results comparable. There are seven cases of defect depths: 6 mm, 7 mm, 8 mm, 9 mm, 10 mm, 11 mm and 12 mm. When the initial phase of each frequency component of the excitation signal is the optimal phase (the proposed IPECD method), the worst phase and 0 (the traditional FSPECT method), the differential pickup signals of the magnetic induction intensity for the different defect depths are calculated, and they are shown in Figure 4a–c, respectively.

For defects with different depths, we define the peak value of the differential pickup signals of the magnetic induction intensity as the detection sensitivity. According to the above definition, the detection sensitivities when using the excitation signals of the different initial phases are compared in Figure 4d. Clearly, the detection sensitivities when using the excitation signals of the different initial phases increase with the decreasing depth of the defects, which is due to the skin depth effects. Additionally, the detection sensitivity of the proposed IPECD method is about 25% higher than that of the traditional FSPECT method, which demonstrates the superiority of the IPECD method over the FSPECT method.

The IPECD method can effectively elevate the maximum amplitude of the excitation signal over the FSPECT method by controlling and optimizing the initial phase of each frequency component, which can further improve the detection sensitivity. Hence, to detect defects of a known depth range, the proposed IPECD method has the advantage of detection sensitivity over the traditional FSPECT method.

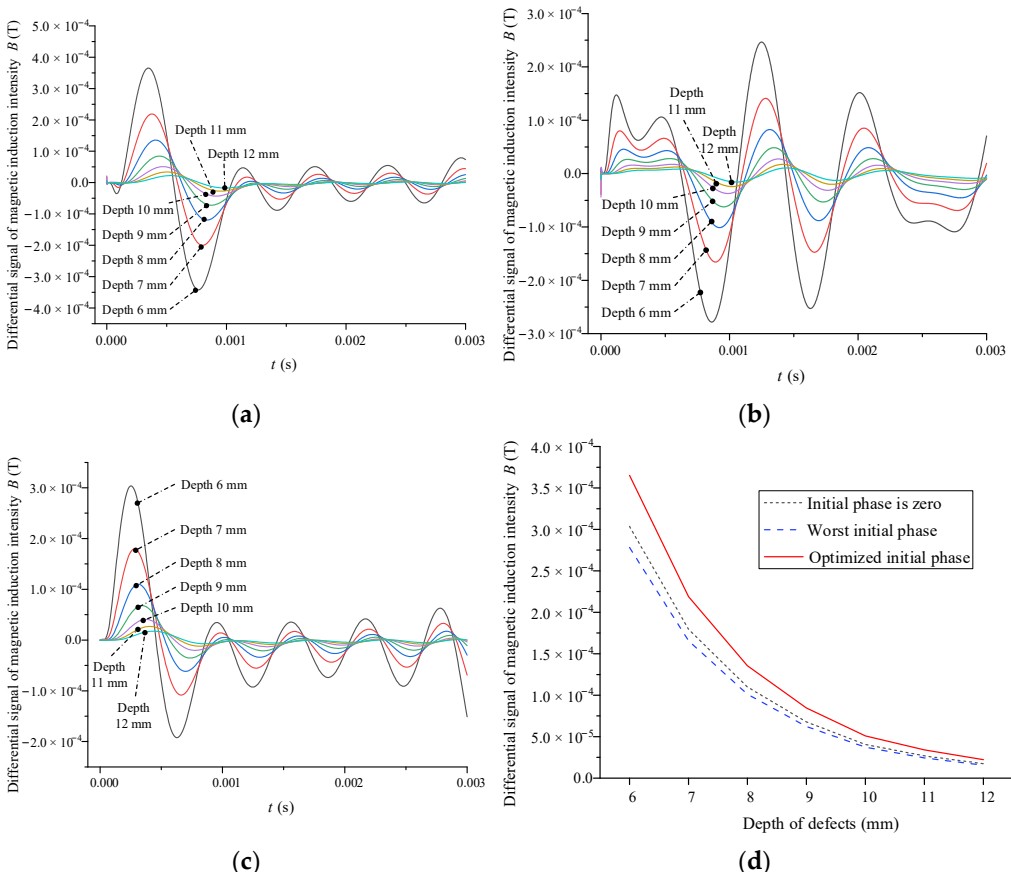

**Figure 4.** Differential detection simulation signal of magnetic induction intensity when the phase angle of excitation signal is (**a**) optimized, (**b**) the worst and (**c**) zero; (**d**) peak values of the differential pickup signals of magnetic induction intensity under different excitation signals.

## 4. Experimental Verification and Result Analysis

In this work, an experimental platform for the deep defect detection of cable lead sealings is built, and by conducting experiments using the IPECD method, the effects of the defect depth on features of the lift-off point of intersection (LOI) and zero-crossing time (ZCT) are studied, which gives references for the prediction or rapid detection of the depth of lead sealing defects.

### 4.1. Experimental Setup

The experimental platform for the defect detection of lead sealings mainly includes a controlling computer, a signal generator, a data acquisition module, an eddy current probe and a scanning platform, as shown in Figure 5. The signal generator generates the excitation signals with optimized initial phases (as shown in Figure 2) with the help of LabVIEW software (Version 2017, National Instruments, Austin, TX, USA). Moreover, the eddy current probe is composed of an excitation coil and a linear Hall sensor SS495A (Honeywell International, Charlotte, NC, USA). The inner/outer diameter, height, wire diameter and number of turns of the excitation coil are 6/10 mm, 5 mm, 0.1 mm and 300, respectively. Additionally, the data acquisition module NI PCI-6256 (National Instruments, Austin, TX, USA) is used to receive the detection signals acquired by the probe. In addition, the controlling computer is used to control the motion vector of the scanning platform and the data acquisition of the data acquisition module.

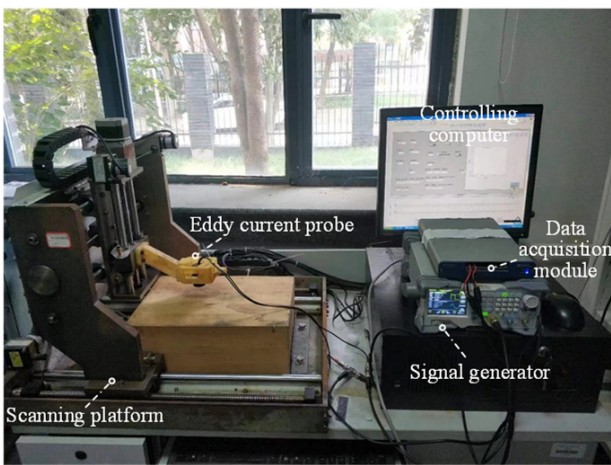

**Figure 5.** Experimental platform for defect detection of lead sealing.

Four lead tin alloy experimental specimens with groove defects across their bottom sides are employed in this work. Moreover, these specimens are of the same sizes and material parameters as the ones described in the numerical model in Section 3.1. Additionally, specimens #1–#4 have bottom defects with 20 mm constant widths and the heights of 8 mm, 10 mm, 12 mm, and 14 mm, respectively, which are used to simulate the corresponding lead sealing defects with buried depths of 12 mm, 10 mm, 8 mm and 6 mm.

*4.2. Results and Discussions*

4.2.1. Effects of the Defect Depth on Features of ZCT

Generally, the zero-crossing phenomenon exists in the pulsed eddy current differential signal of non-ferromagnetic materials, and ZCT is considered an important parameter to reflect the defect characteristics in the time domain [16]. Interestingly, in this work, we also find that such a promising signal feature also exists in the differential pickup signals of the magnetic induction intensity when detecting the defects in the lead tin alloy specimens using the IPECD method. Thus, by conducting experiments using the IPECD method, we explore the effects of the defect depth on the features of ZCT.

In this section, there are four cases of depth defects (6 mm, 8 mm, 10 mm and 12 mm), and the lift-off distance between the probe and the specimen is fixed at 0.5 mm. Moreover, the differential pickup signals of the magnetic induction intensity under the different defect depths are obtained experimentally and shown in Figure 6a,b. Obviously, there is always a zero-crossing phenomenon in the differential pickup signals of the magnetic induction intensity.

After the existence of zero-crossing phenomenon in the differential pickup signals is verified by the experiments, the effects of the defect depth on features of ZCT is further studied using the IPECD method, as shown in Figure 6c. Here, the time of the first zero crossing of the differential pickup signals is defined as the characteristic value of ZCT. It can clearly be concluded that the value of ZCT increases with an increase in the defect depth. Moreover, the relationship between ZCT and the defect depth $D$ can be further described using Equation (6). The relationship described in Equation (6) suggests that, when only the defect depth of the lead sealing is changed, ZCT of the differential pickup signals can serve as a critical parameter to predict or rapidly detect the depth of the lead sealing defect in a practical inspection.

$$ZCT = 3.361 \times 10^{-4} + 3.66 \times 10^{-5}D \tag{6}$$

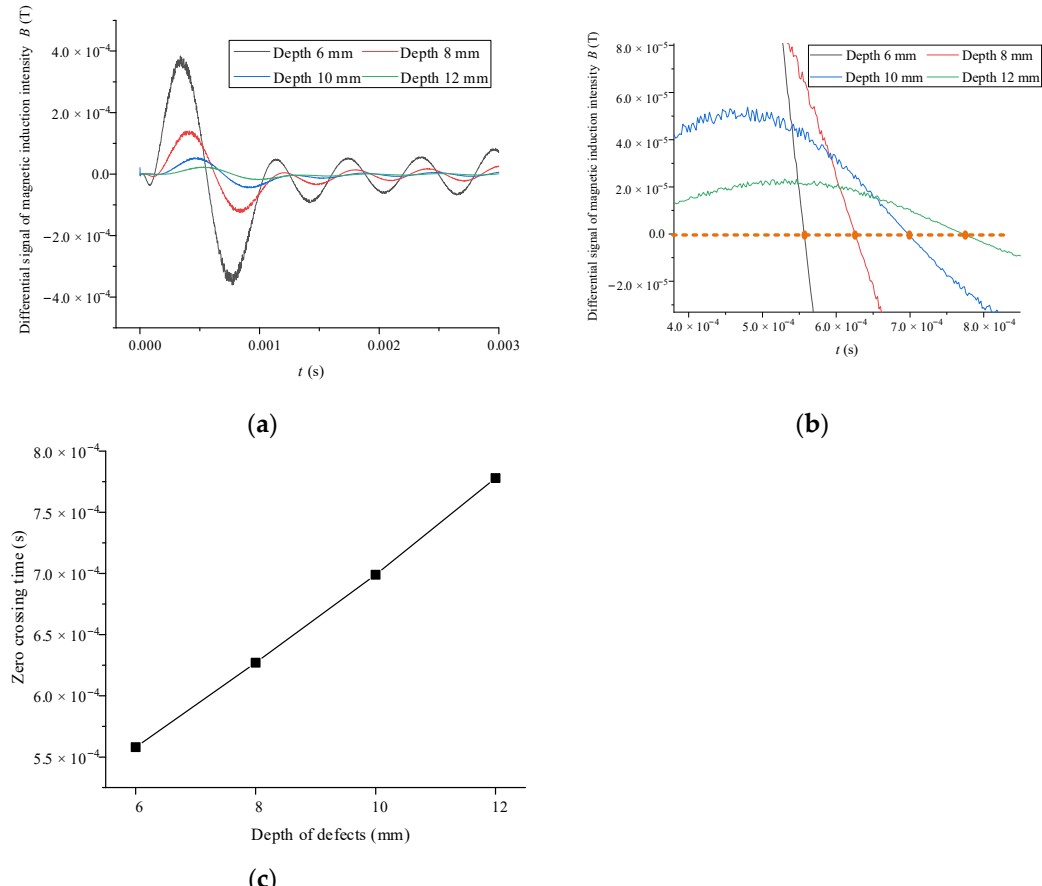

**Figure 6.** (**a**) Differential pickup signals of magnetic induction intensity under different defect depths; (**b**) ZCT of differential pickup signal under different defect depths; (**c**) effects of the defect depth on ZCT.

4.2.2. Effects of the Defect Depth on Features of LOI

Generally, LOI, as a signal feature immune to the change in the lift-off distance, can effectively inhibit the influence of non-conductive coating or surface contamination on a pulsed eddy current detection signal [17], which inspires us that the above promising signal feature may also exist in the differential pickup signals detected using the IPECD method. Hence, we analyze the characteristics of the differential pickup signals in detail when using the IPECD method and, by conducting experiments, further explore the effects of the defect depth on features of LOI.

In this section, there are four cases of defect depths (6 mm, 8 mm, 10 mm and 12 mm), and there are three cases of lift-off distances between the probe and the specimen (0.5 mm, 1.0 mm, and 1.5 mm) at each defect depth. Moreover, the differential pickup signals of the magnetic induction intensity under different defect depths are obtained experimentally, and they are shown in Figure 7a–d. It can be clearly observed that, when only the lift-off distance varies, the differential pickup signals intersect at a point, which is defined as the LOI that can eliminate noise interference caused by lift-off distance.

To quantify the defect depth using the LOI feature, we analyze the effects of the defect depth on LOI features. Here, the time corresponding to the LOI of the differential pickup signals is used as the time characteristic value of LOI, which is defined as the time of LOI. The relationships between the time of LOI and the defect depth are shown in Figure 7e. Obviously, when the defect depth increases, the time of LOI increases; i.e., the LOI feature comes later. Moreover, the relationship between the time of LOI and the defect depth $D$ can be further described using Equation (7). Hence, in a practical inspection of lead sealing

defects, the critical parameter of the time of LOI can be acquired by varying the lift-off distance. Then, the defect depth of the lead sealing can be predicted or rapidly detected.

$$\text{Time of LOI} = 3.56 \times 10^{-4} + 3.1 \times 10^{-5}D \tag{7}$$

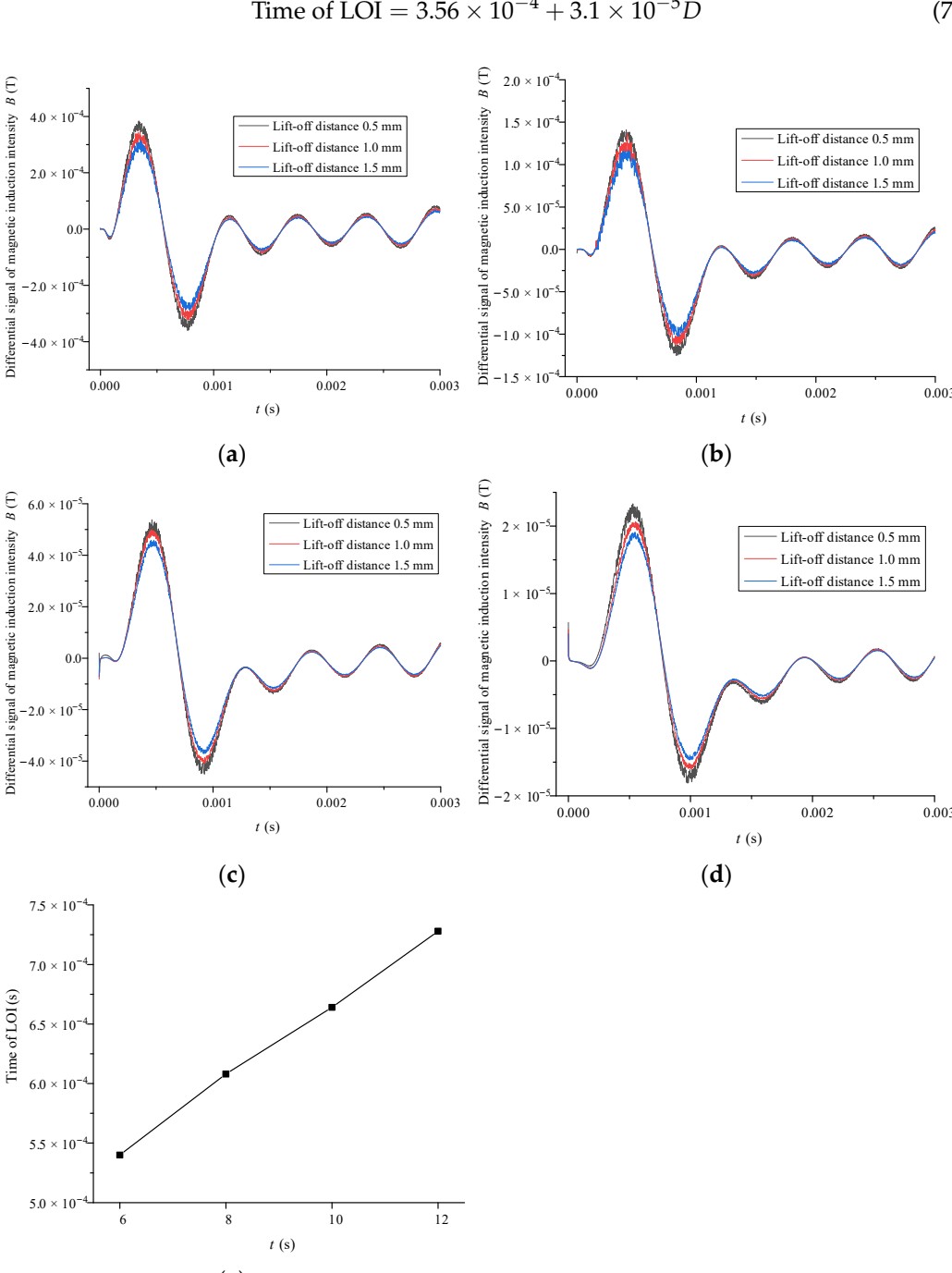

**Figure 7.** Differential pickup signals of magnetic induction intensity under different lift-off distances: (**a**) 6 mm defect depth, (**b**) 8 mm defect depth, (**c**) 10 mm defect depth, (**d**) 12 mm defect depth; (**e**) variations in time of LOI with defect depths.

## 5. Conclusions

In this work, an IPECD method is proposed to detect the defects of lead sealings ranging from 6 to 12 mm. First, the frequency range selection principle and the optimization method of initial phase angles of different frequency components are explained in detail. Second, simulated investigations on the detection sensitivities of the defects before and

after the initial phase angle optimization are carried out. The results show that the detection sensitivity of the proposed IPECD method (where the initial phases are optimized) is about 25% higher than that of the traditional FSPECT method (where the initial phases are zero). Finally, an experimental system is built to study the effects of the defect depth on the features of LOI and ZCT. The results indicate that there is always a zero-crossing phenomenon in the differential pickup signals of the magnetic induction intensity, and the value of ZCT basically increases linearly with an in increase in the defect depth. Additionally, when only the lift-off distance varies, the differential pickup signals intersect at a point. Moreover, the time of LOI also basically increases linearly with an increase in the defect depth.

**Author Contributions:** Conceptualization, Q.S. and S.F.; methodology, Q.S.; software, Q.S.; validation, Q.S. and S.F.; formal analysis, F.L.; investigation, Q.S.; resources, Q.S.; data curation, Q.S.; writing—original draft preparation, Q.S.; writing—review and editing, Q.S. and S.F.; visualization, Q.S.; supervision, Q.S.; project administration, Q.S.; funding acquisition, Q.S. and S.F. All authors have read and agreed to the published version of the manuscript.

**Funding:** This research was funded by the Key Scientific and Technical Funds of Sichuan Electric Power Corporation, grant number 52199722000R.

**Conflicts of Interest:** The authors declare no conflict of interest.

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
