# Peer review of "Research on Deep Defect Detection Method of Cable Lead Sealing Based on Improved Pulsed Eddy Current Excitation"

_electronics, doi:10.3390/electronics11152428_

Round 1
Reviewer 1 Report
In the manuscript the PEC method to detect defects of cable lead sealing has been presented. This approach is an improvement on the traditional FSPECT method, which achieves the detection sensitivity higher by 25%.
I appreciate the improvement presented, but I find the novelty of the paper relatively low. Furthermore, the manuscript was not written in scientific and systematic ways, for example there is not even a table with the numerical values which makes it very difficult to verify the results obtained.
My comments are included below:
11) What does this sentence mean: “in this work, an improved pulse eddy current detection (IPECD) method to detect the deep defects of cable lead sealing (depth ranges from 6 to 12 mm) is introduced,”? In my opinion, such nomenclature may be confusing to the reader and may suggest that the PECD method has already been published earlier and that an improved version of it (IPECD) has now been presented.
22) “Due to the small eddy current skin depth caused by the high-frequency characteristics of alternating current and the waste of frequency caused by the wide-frequency characteristics of square wave pulse, the current limit depth of lead sealing defect detection is only 6 mm.” Eddy current skin depth depends inversely on the frequency used. Explain why it is not possible to use an AC pulse at a lower frequency to obtain a greater depth of defect detection than 6 mm with the classic method?
33) Should expression (1) not include the permeability of free space m0? The permeability of the medium is the product of the relative permeability of the medium (1 for lead sealing) and the permeability of free space.
44) Typos:
Page 3: “and the ith selected frequency can be expressed as:”. Correct “ith”.
Page 5: “The depths of the defects have 7 cases, 6 mm, 7 mm, 8 mm, 9 mm, 10.0 mm, 11 mm and 12 mm.”. Replace 10.0 mm with 10 mm.
55) Page 3: “are calculated by genetic algorithm,”. What exactly was the algorithm used?
66) I do not see information about the parameters of the FEM model. What package was used to perform the simulation? How many elements did the mesh have? What was the shape of the elements?
77) „In order to avoid the influence of the size of the air domain on the simulation accuracy, the diameter and the height of the air domain are set to 150 mm and 60 mm, respectively.” This formulation needs to be clarified. Increasing the solution domain disturbs the correct results? Shouldn't it be that the larger the solution domain the smaller the error? What boundary conditions were used in the simulation?
88) The figures presented in the paper are of poor quality. For experimental platform (Fig. 5), a photo of the stand should be provided.
99) There is the same text in three places in the manuscript!
Introduction:
“Consequently, in this work, an improved pulse eddy current detection (IPECD) method to detect the deep defects of cable lead sealing (depth ranges from 6 to 12 mm) is introduced, and the frequency range selection principle and the optimization method of initial phase angles of different frequency components of IPECD to maximize the peak value of the excitation signal under the same input power is explained in details at first. Then, the detection sensitivities of the lead sealing deep defect before and after the initial phase angle optimization of different frequency components are compared and analyzed based on the simulation. Finally, the effects of the defect depth on features of lift-off point of intersection (LOI) and zero-crossing time (ZCT) are studied using the IPECD method by experiments, which enhances the foundation for the prediction or rapid detection of the depth of lead sealing defects.”
Conclusions:
“In this work, an improved pulse eddy current detection (IPECD) method to detect the deep defects of cable lead sealing (depth ranges from 6 to 12 mm) is introduced, and the frequency range selection principle and the optimization method of initial phase angles of different frequency components of IPECD to maximize the peak value of the excitation signal under the same input power is explained in details at first. Then, the detection sensitivities of the lead sealing deep defect before and after the initial phase angle optimization of different frequency components are compared and analyzed based on the simulation. … Finally, the effects of the defect depth on features of lift-off point of intersection (LOI) and zero-crossing time (ZCT) are studied using the IPECD method by experiments. “
Abstract:
“… an improved pulse eddy current detection (IPECD) method to detect the deep defects of cable lead sealing (depth ranges from 6 to 12 mm) is introduced, and the frequency range selection principle and the optimization method of initial phase angles of different frequency components of IPECD to maximize the peak value of the excitation signal under the same input power is explained in details at first. Then, the detection sensitivities of the lead sealing deep defect before and after the initial phase angle optimization of different frequency components are compared and analyzed based on the simulation. Finally, the effects of the defect depth on features of lift-off point of intersection and zero-crossing time are studied using the IPECD method by experiments, which enhances the foundation for the prediction or rapid detection of the depth of lead sealing defects.”

Reviewer 2 Report
The work is concise and clearly written. The problem for me is references that are exclusively written in Chinese. I couldn't get to most of the articles. Personally, I do not see much novelty of the work in terms of the new method except for the abbreviation. The paper is clear, but I miss the description of the measurement setup because it represents the author's contribution in terms of the research. Possibilities for improvement are:
In chapter ''2. Basic principle of the IPECD method'' third row from the bottom '' 4.67 × 106 S/m'' propose is to not use the vector product but the multiplication sign. In the second chapter, in addition to the mentioned frequencies [300 Hz, 1500 Hz], it would be nice to stated which frequency correspond to which penetration gain. Figure 2 is readable, but it would be good to see a vector chart instead of a bitmap if possible. Figure 3 would be more transparent if the sizes in mm were quoted directly on the model, it is a suggestion, but what should be done is to put the length of mm next to the axis, because it is not noticeable due to overlapping with the lines of the model. In Figure 4, the values should be separated from the unit (5mm => 5 mm); Depth may be written only once. It would be good to put a photo of the measuring setup in chapter 4.1. Experimental setup
Round 2
Reviewer 1 Report
All my previous comments have been considered and sufficiently implemented.